# ES-Parkour: Advanced Robot Parkour with Bio-Inspired Event Camera and Spiking Neural Network

## Abstract

In recent years, significant progress has been made in the field of quadruped robotics, particularly in perception and motion control algorithms powered by reinforcement learning. It has shown that these robots are capable of executing complex motions in challenging environments with the help of visual sensors like depth cameras, demonstrating outstanding stability and robustness. However, these sensors typically operate at lower frequencies compared to the control frequency of the robot's joints and are susceptible to lighting conditions, making them challenging to deploy outdoors. Moreover, the computational load on the robot's side is increased by the deep neural networks used in multiple sensor systems and control units. To address these challenges, we, *for the first time*, introduce the spiking neural networks (SNNs) and event cameras to accomplish a complex quadruped robot parkour task. This combination leverages the efficiency of SNN in processing spike sequences and the capability of event cameras to capture dynamic visual information, thus showing great potential in emulating biological perception and processing mechanisms. Our experimental results indicate that employing event cameras and SNN yields excellent performance in challenging parkour tasks. Compared to traditional deep neural networks, our ES-Parkour presents significantly lower energy consumption, amounting to merely **11.7%** of that exhibited by the ANN model. This corresponds to an extreme energy-saving **(88.3%)** by utilizing SNN. By integrating the strengths of event cameras and SNN, our work expands the possibilities for the further development of robotic reinforcement learning algorithms and explores their future applications in various challenging environments.

Quadruped robots have made significant strides in recent years, not only in motion control relying on proprioception (Reske et al., 2021; Hwangbo et al., 2019; Wu et al., 2023; Peng et al., 2020; Iscen et al., 2018) but also in planning control relying on various visual sensors. Current quadruped robots can accomplish a variety of tasks in many different complex environments (Zhuang et al., 2023; Cheng et al., 2023). They have demonstrated enormous application potential and can assist humans in working in various extreme environments.

Our goal is to enable quadruped robots to function in extremely challenging conditions, including diverse lighting environments, highly complex scenarios, and long-endurance situations. Despite significant advancements in motion control and perception, research addressing these specific challenges remains scarce. We draw inspiration from the work of Shi, L. et al. (Yu et al., 2023), who employed brain-inspired algorithms and sensors to control quadruped robots in various settings, showcasing the potential of brain-inspired neural networks and event cameras. However, their perceptual environment was relatively simplistic. Building upon their research, we embark on a comprehensive study. We adopt the Parkour task as a benchmark for the perceptual and motion capabilities of quadruped robots, develop a corresponding simulation environment, and establish rigorous experimental conditions for testing.

Research on brain-inspired neural networks and devices is currently a cutting-edge direction. Through brain-inspired design, we can achieve neural networks and perception sensors with lower power consumption and higher frequency. These advantages are crucial for robots. Robots need high-frequency perception to model the environment, and the control frequency of robot joints is generally extremely high. However, the inference speed of current neural networks is often much lower than the robot joint control cycle. Therefore,

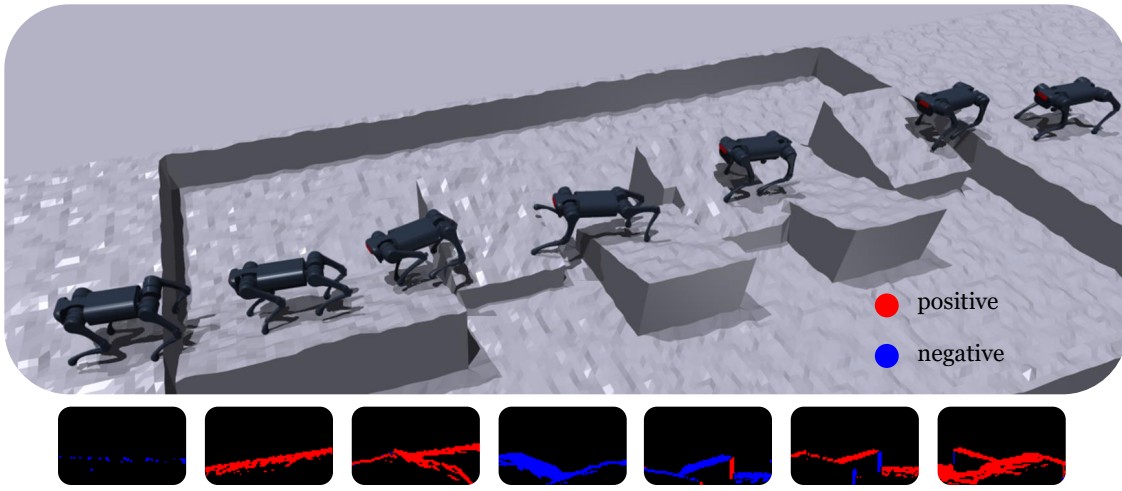

Figure 1: **Demonstration of a quadruped robot performing parkour using spiking neural network under extreme lighting conditions.** The robot processes event images in real-time, where the red part and blue part denote the positive event and negative event, respectively.

we explore the improvements and assistance that brain-inspired devices and neural networks can provide for quadruped robots in various motion scenarios. Moreover, by employing more biomimetic devices and networks, we effectively control power consumption. This aspect is crucial for the structural design of robots. Currently, many robot designs have to compromise between mechanical structures and electronic devices due to issues with power consumption, especially in terms of incorporating numerous active cooling devices and batteries. Our approach can alleviate this problem to a certain extent.

Event camera is a novel type of visual sensor that can output pixel-level changes in the environment at very high frequencies. The output of these sensors is independent of light intensity, allowing for their use in outdoor scenarios. Additionally, the high output frequency of these sensors, reaching up to several kilohertz, can match the control frequency of robots, thereby enhancing the potential of existing reinforcement learning algorithms. Spiking Neural Networks (SNN) represent a new class of neural networks whose inputs and outputs are spike signals. These networks can significantly reduce the number of parameters without loss of accuracy, thus decreasing computational requirements. SNN are inspired by the way biological neurons work: they fire or "spike" only when a membrane potential, an electric potential difference across the cell membrane, reaches a specific value. This makes them highly efficient in terms of power and computational resources, as they only need to process data when the neurons fire.

In our work, we employ an event camera as the only visual sensor. We found that the event camera's representation of object contours significantly enhances the performance of complex perception tasks for quadruped robots. Previous studies often required 3D sensors such as depth cameras or LiDAR, or even the design of a multi-sensor system, to accomplish complex perception tasks. In contrast, our study leverages the potential of the event camera, a 2D sensor, to perceive complex environments. Similar to previous work on drones (Kaufmann et al., 2023; Falanga et al., 2020), we found that the event camera's contour description characteristic is highly suitable for motion control of quadruped robots. Furthermore, we utilize SNN, which is often used in conjunction with event cameras. The use of SNN has significantly reduced our computational load. This efficiency is particularly beneficial in our work, as it allows us to perform complex perception tasks with less computational power, making our approach more feasible for real-world applications in the future. The whole pipeline of our proposed bio-inspired system is illustrated in Figure 2.

Our work makes three primary contributions:

1. We pioneer the implementation of a system-level design for robot parkour using Spiking Neural Networks (SNNs) and event cameras. The experimental results provide new perspectives and potential for the perception and motion control of quadruped robots.

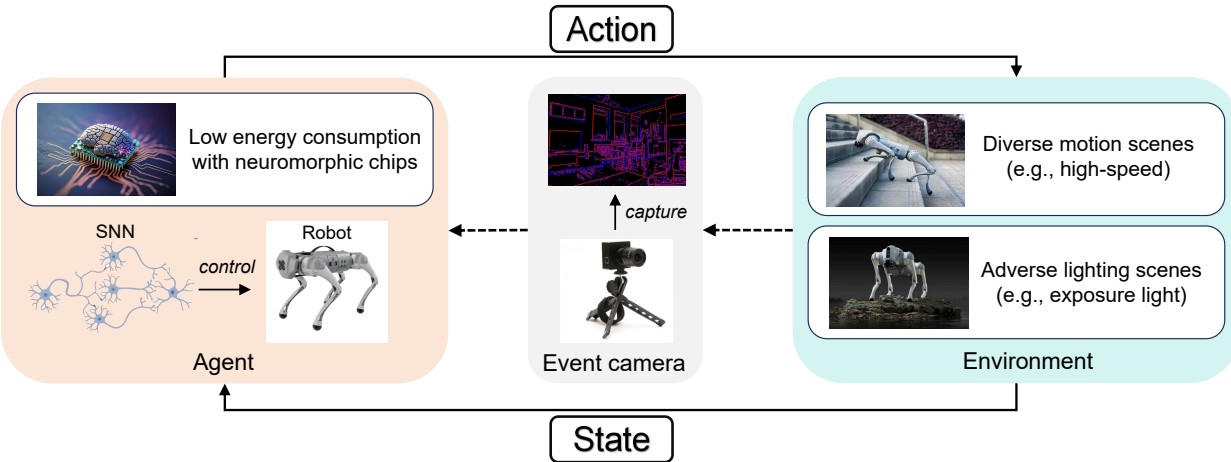

Figure 2: **Pipeline of our bio-inspired reinforcement learning system.** Different from the previous standard vision-based robot system, our bio-inspired system is equipped with an event camera to capture event data from diverse scenes. The event is then processed by the spiking neural network which in turn dictates the robot's actions to the environment. The adoption of this brain-inspired approach yields three significant advantages: (1) enhanced stability in motion-intensive scenarios is achieved through the superior temporal resolution of the event data. (2) the system's resilience in fluctuating lighting conditions is ensured by the event camera's high dynamic range. (3) the inherently low energy consumption of the SNN contributes to the system's overall efficiency.

2. We successfully transition the end-to-end training of the quadruped robot reinforcement learning network from artificial neural networks to spiking neural networks using the distillation method. This transition significantly reduces the computational burden, training difficulty, and training cost, enhancing the practicality and feasibility of our approach.

3. To our knowledge, this is the first demonstration of achieving complex quadruped robot control tasks across a variety of environments using a 2D brain-inspired sensor, supported by extensive testing in diverse settings. Our work ensures that robots possess robust perception and control capabilities in various environments. The experimental results demonstrate that event cameras can effectively manage the extreme movements of quadruped robots, significantly broadening the application scope of brain-inspired devices in robots.

Although breakthrough research has already been conducted in drones (Davide et al., 2020) and autonomous driving (Gehrig & Scaramuzza, 2024), the application of event cameras in legged robots remains limited. This scarcity is primarily due to the complexities involved in designing and implementing the systems. However, the combination of event cameras and spiking neural networks (SNN) holds significant potential for the robotics community. We hope our extensive simulation validation work will pave the way for further advancements in this field.

# 1 Related Work

## 1.1 Legged Robot Agile Locomotion

Due to the flexibility and stability of Legged robots, many researchers have forced on agile locomotion controllers. The methods are primarily categorized into optimization-based (Bledt & Kim, 2020; Di Carlo et al., 2018; Grandia et al., 2019a; Ding et al., 2019; Kim et al., 2023) and learning-based approaches (Reske et al., 2021; Hwangbo et al., 2019; Lee et al., 2020; Wu et al., 2023; Peng et al., 2020; Iscen et al., 2018). Trajectory optimization (TO) (De Viragh et al., 2019; Chignoli & Kim, 2021; Cebe et al., 2021), is a prevalent form of optimal control used for creating motion patterns in legged systems. These methods

generate dynamically stable base trajectories of quadruped robots for whole-body control (Bellicoso et al., 2017). Model Predictive Control (MPC) is also increasingly popular in robotics for managing the motion of complex and dynamic systems. Its capacity to handle non-linearities and constraints makes it a preferred method in the legged robot field (Gaertner et al., 2021; Grandia et al., 2019b; Ding et al., 2019). However, these approaches usually need a real-time model of the robot's kinematics and dynamics and a detailed representation of the terrain, which need multi-sensors and accurate state estimation. In contrast to the optimization-based methods, the learning-based methods don't need complex modeling processes for robots and terrain dynamics. Rapid Motor Adaptation (RMA) (Kumar et al., 2021) introduces a teacher-student framework to enhance the student policy ability across the challenging terrain by utilizing the privileged-aware teacher policy. (Wu et al., 2023) combines the RMA and Adversarial Motion Priors (AMP) (Peng et al., 2021) to generalize the policy from the reference dataset on flat terrains to complex terrains in real-world scenarios. (Li et al., 2023) propose a cooperative adversarial framework for learning multiple motion skills from unlabeled datasets. Moreover, the learning-based methods also can directly process information from visual sensors through neural networks without precise mapping. Recent studies in the vision-guided robotics locomotion domain have shown significant advancements. For instance, (Imai et al., 2022) contact proprioceptive and vision features obtained from depth sensors to enhance environmental perception. (Yang et al., 2023) innovatively encoded historical views into 3D volume features, combining them with the robot's proprioceptive state for a more comprehensive 3D environmental understanding. Moreover, (Hoeller et al., 2022) proposed a learning-based method to reconstruct the local terrain for locomotion tasks. This technique is further utilized by (Duan et al., 2023; Bellegarda & Ijspeert, 2022; Hoeller et al., 2023) utilize this similar method to generate the local map for policy.

## 1.2 Robotic Parkour

Robotic parkour represents a rapidly emerging and challenging domain in robotics, demanding highly effective and accurate algorithms to enable robots to navigate complex and high-risk terrains successfully. (Zhuang et al., 2023) a novel approach by pre-training parkour task skills under soft constraints in varied scenarios. This method is further refined by fine-tuning the model under more stringent, hard constraints and distilling multiple skills into a singular policy framework. (Cheng et al., 2023) merges two perception approaches within the teacher-student framework. They obtain local height maps and target directions from the simulator for teacher policy pre-train and distill this privileged knowledge into student policy which only uses depth images. (Hoeller et al., 2023) employed a hierarchical framework that integrates a navigation policy with multiple motion policies, enabling effective traversal over challenging terrain. However, these mentioned approaches only use traditional vision sensors such as depth cameras or lidar, which results in poor performance in extreme lighting conditions and fast scenes.

## 1.3 Event Camera and SNN on robots

The research related to brain-inspired algorithms (Tang et al., 2021; Florian, 2007; O'Brien & Srinivasa, 2013; Frémaux et al., 2013) and devices (Mahlknecht et al., 2022; Zhu et al., 2018; Monforte et al., 2023) in robotics is promising. (Zihao Zhu et al., 2017; Chen et al., 2023; Guan et al., 2023) explore the robotics application of event-based cameras on Visual Inertial Odometry (VIO) or Simultaneous Localization and Mapping (SLAM). Due to the ability of event cameras to capture fast dynamic scenes, these works apply them in location and navigation modules of robots (especially drones) (Forrai et al., 2023) equipped their quadruped with an event camera to catch the high-speed ball. However, these mentioned methods still utilize ANN or traditional approaches to deal with event data rather than SNN. SNN are particularly well-suited for event cameras due to their temporal precision. Event cameras capture data asynchronously, only when changes in brightness are detected. SNN can process this data in a time-resolved manner, making them ideal for capturing the dynamic and temporal aspects of the visual scene. (Tang et al., 2021) applies a population-coded spiking actor network and a deep artificial critic network to achieve continuous control tasks in simulation. (Jiang et al., 2023) implement SNN in legged robots simulation environments. Due to the limitations of the physical simulator, these models are incapable of obtaining and dealing with event-based data. (Yu et al., 2023) integrates multimodal cues from traditional and bio-inspired sensors with SNN and

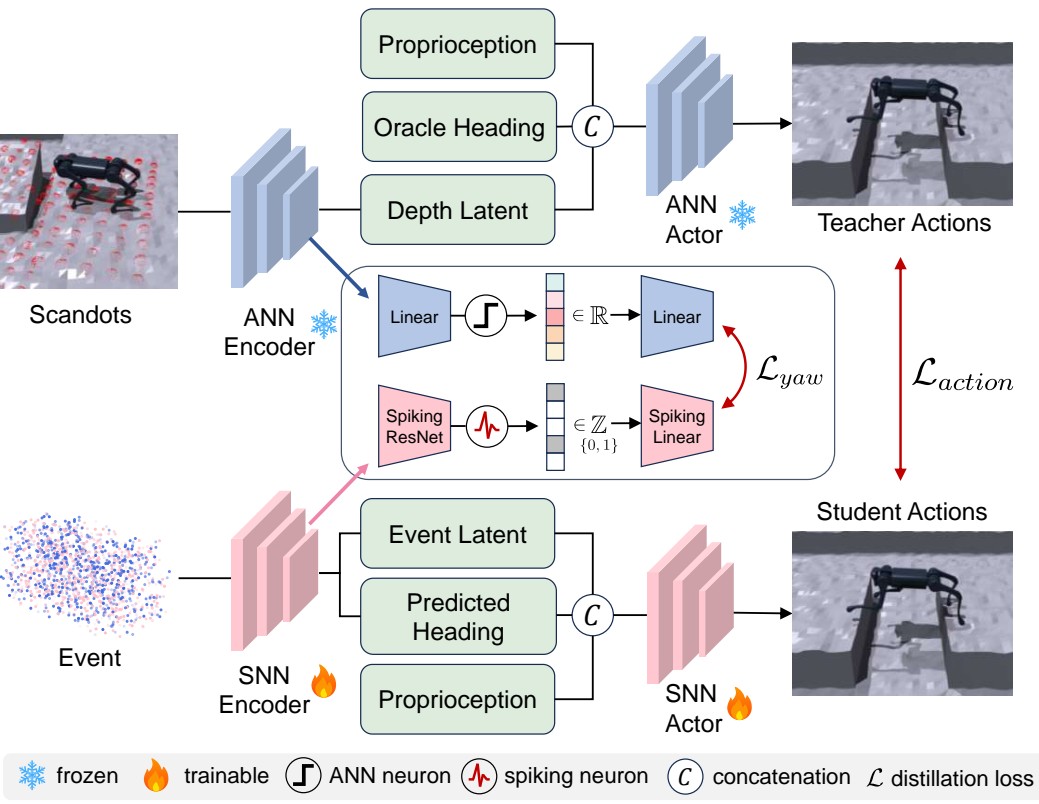

Figure 3: **Pipeline of our ES-Parkour ANN-to-SNN distilling process.** Through the distillation process, the extreme parkour capabilities of the ANN are transferred to an SNN, which receives input from an event camera. In the warm-up phase, minimizing the Mean Squared Error (MSE) loss between the outputs of the teacher (ANN) and the student (SNN) networks ensures the student network can closely replicate the teacher network's outputs. Following the warm-up phase, the student network demonstrates basic movement capabilities but encounters challenges with complex terrains. Further interaction and optimization of the student network enhance its performance on complex terrains, closely aligning it with the teacher network's performance.

ANN. However, these mentioned methods do not integrate robot perception and control into one framework using SNN.

## 2 Method

### 2.1 Motivation

Depth cameras are one of the most commonly used sensors in robotics due to their important properties of detecting the presence of any object nearby and measuring the distances (Cheng et al., 2023; Sefercik & Akgun, 2023). Complementarily, RGB cameras furnish a rich tapestry of pixel information, augmenting scene recognition capabilities across diverse environments, and can be effectively integrated with depth cameras for enhanced sensorial input (Liu et al., 2022; Xu et al., 2023). However, in real environments, robots operating in real-world settings frequently meet a wide range of lighting scenarios (e.g., extreme exposure or darkness), and encounter diverse motion scenes (e.g., high-speed or low-speed). Traditional RGB cameras or depth cameras are incapable of capturing adequate information in bright or dark light situations and often lead to blurry or distorted results in motion scenes, as detailed in Table 1. To address the above issues, we choose to use event cameras as our sensors, thereby ensuring robust and precise environmental interaction and knowledge acquisition.

Moreover, directly implementing complex convolutional or transformer neural networks on robots is significantly challenging due to the rapidly expanding size of visual models. Most ANN rely on GPU chips or specialized acceleration chips for inference, leading to power consumption becoming a major concern in the design of robotic structures. Through discussions with numerous experts in robot structural design, we've identified that optimizing the active cooling systems, such as fans required by these high-power chips within the compact framework of quadruped robots, is particularly difficult. This challenge has spurred our interest in exploring neural networks and chips that consume less power. SNN, when used alongside event cameras, has captured our focus. SNN and their corresponding chips can drastically reduce power consumption while still maintaining efficient information processing capabilities. This feature makes SNN an excellent option for fulfilling the real-time perception and decision-making requirements of robots in complex environments.

The bio-inspired attributes of SNN extend beyond just low power consumption, they also include efficient processing of event-driven information. This complements the way event cameras detect changes in the environment, making their combination particularly effective for visual perception tasks. As a result, SNN can adeptly manage these sparse yet highly informative data sets, further reducing computational complexity and power usage.

## 2.2  Build Event Camera in Simulation

Event Camera are bio-inspired sensors, which capture the relative intensity changes asynchronously. In contrast to standard cameras that output 2D images, event cameras output sparse event streams. When brightness change exceeds a threshold $C$, an event $e_k$ is generated containing position $\mathbf{u} = (x, y)$, time $t_k$, and polarity $p_k$:

$$\Delta L(\mathbf{u}, t_k) = L(\mathbf{u}, t_k) - L(\mathbf{u}, t_k - \Delta t_k) = p_k C. \qquad (1)$$

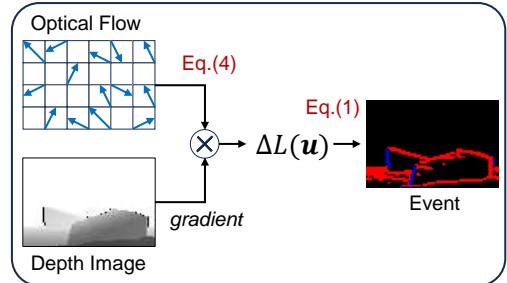

The polarity of an event reflects the direction of the changes (*i.e.*, brightness increase ("ON") or decrease ("OFF")). In general, the output of an event camera is a sequence of events, which can be described as: $\mathcal{E} = \{e_k\}_{k=1}^{N} = \{[\mathbf{u}_k, t_k, p_k]\}_{k=1}^{N}$, where $e_k$ is the $k$-th event, $(x_i, y_i)$ is the pixel location, $t_i$ records the timestamp, $p_i$ denotes polarity.

Figure 4: **Overview of the event simulation process.** Each depth image can be converted into its corresponding event with the optical flow and image gradient.

In this paper, we utilize IsaacGym as the simulation and training environment. IsaacGym is a high-performance robotic simulation platform that provides a rich physical simulation environment, enabling us to efficiently train and test quadruped robots in complex scenarios. However, the IsaacGym platform does not natively support the simulation of event cameras (Event Camera). Therefore, we develop an algorithm to simulate the working principle of event cameras within the IsaacGym environment:

Suppose that in a small time interval, the brightness consistency assumption (Horn & Schunck, 1981) is conformed, under which the intensity change in a vicinity region remains the same. By using Taylor's expansion, we can approximate intensity change by:

$$\Delta L(\mathbf{u}, t) = L(\mathbf{u}, t) - L(\mathbf{u}, t - \Delta t), \qquad (2)$$

$$= \frac{\delta L}{\delta t}(\mathbf{u}, t)\Delta t + O(\Delta t^2) \approx \frac{\delta L}{\delta t}(\mathbf{u}, t)\Delta t, \qquad (3)$$

where $\mathbf{u} = (x, y)$ denotes the position. Substituting the brightness constancy assumption ($\frac{\delta L}{\delta t}(\mathbf{u}(t), t) + \nabla L(\mathbf{u}(t), t) \cdot \mathbf{v}(\mathbf{u})) = 0$.) into the above equation, we can obtain:

$$\Delta L(\mathbf{u}) \approx -\nabla L(\mathbf{u}) \cdot \mathbf{v}(\mathbf{u})\Delta t, \qquad (4)$$

which indicates that the brightness changes are caused by intensity gradients $\Delta L = (\frac{\delta L}{\delta x}, \frac{\delta L}{\delta y})$ moving with velocity $\mathbf{v}(\mathbf{u})$ over a displacement $\Delta \mathbf{u} = \mathbf{v}\Delta t$. As expressed by the dot product in Eq. 4, if the moving

| Camera Type | RGB Camera | Depth Camera | Event Camera |
|---|---|---|---|
| Dynamic Range | Low (∼60dB) | Low | High ( $\geq$ 120dB ) |
| Latency | High | High | Low |
| Advantages | • Versatile for many conditions 
 • High color fidelity | • Captures spatial data for 3D modeling 
 • Useful in AR/VR | • Great for capturing movement in high dynamic range scenes |
| Disadvantages | • Limited in extreme lighting without HDR | • Limited functionality in diverse light conditions | • Less effective for static scenes |

Table 1: **Comparison of RGB, Depth, and Event Cameras.** Event cameras, with their high dynamic range (HDR) and low latency, are ideally suited for robotic applications in outdoor and extreme-exposure environments.

direction is parallel to the brightness gradient (*i.e.*, $\mathbf{v} \perp \nabla L$), no events are generated. With $\mathbf{v}(\mathbf{u})$ and $\nabla L(\mathbf{u})$, we can get $\Delta L(\mathbf{u})$ to generate event data with Eq. 1. In this paper, we adopt the same simulated methods to obtain $\mathbf{v}(\mathbf{u})$ and $\nabla L(\mathbf{u})$ from (Cao et al., 2023), where only a single depth image is required to simulate the corresponding event frames.

Our simulation algorithm calculates pixel changes in the environment in real-time and converts these changes into "events" that would be output by an event camera. Furthermore, we optimize the simulation environment to ensure the real-time performance and accuracy of the event camera simulation. By finely adjusting the simulation parameters, we make the simulated event data as close as possible to the output of a real event camera.

## 2.3 Build SNN in Simulation

The spiking neural network is a bio-inspired algorithm that mimics the actual signaling process occurring in brains. Compared to the artificial neural network, it transmits sparse spikes instead of continuous representations, offering benefits such as low energy consumption and robustness. In this paper, we adopt the widely used Leaky Integrate-and-Fire (LIF (Hunsberger & Eliasmith, 2015)) model, which effectively characterizes the dynamic process of spike generation and can be defined as:

$$U[n] = e^{\frac{1}{\tau}} V[n-1] + I[n], \tag{5}$$

$$S[n] = \Theta(U[n] - \vartheta_{\text{th}}), \tag{6}$$

$$V[n] = U[n](1 - S[n]) + V_{\text{reset}} S[n], \tag{7}$$

where $n$ is the time step, $U[n]$ is the membrane potential before reset, $S[n]$ denotes the output spike which equals 1 when there is a spike and 0 otherwise, $\Theta(x)$ is the Heaviside step function, $V[n]$ represents the membrane potential after triggering a spike. In addition, we use the "hard reset" method (Fang et al., 2021b) for resetting the membrane potential in Eq. 7, which means that the value of the membrane potential $V[n]$ after triggering a spike ($S[n] = 1$) will go back to $V_{\text{reset}} = 0$.

## 2.4 Learning Process

In this section, we present the ANN teacher policy alongside the SNN student training methodology. The ANN teacher networks undergo training utilizing privileged information, encompassing both target direction and terrain scandot data. Following this, we employ a knowledge distillation approach to transfer locomotion insights from the ANN teacher policy to the SNN student policy. Unlike the ANN, which relies on scandot data for terrain perception, the SNN student policy employs depth imaging to navigate complex terrains effectively. In this way, the SNN policy can perceive the environment and compute the target direction only by utilizing the event.

### 2.4.1 Reinforcement Learning on ANN

Our policy training framework is structured as a Markov Decision Process (MDP), defined by the tuple $(\mathcal{S}, \mathcal{A}, \mathcal{R}, p, \gamma)$, where $\mathcal{S}$ denotes the state space, $\mathcal{A}$ represents the action space, $\mathcal{R}$ is the reward function, $p$ characterizes the transition probabilities between states for each action-state pair, $\gamma \in [0, 1]$ is the discount factor applied to rewards. At each time step $t$, the agent receives a state $s_t \in \mathcal{S}$. Based on this observation, the agent selects an action $a_t \in \mathcal{A}$ which is sampled from policy $\pi(a_t|s_t)$. This action leads to a transition $s_t$ to a new state to $s_{t+1}$ determined probabilistically by $s_{t+1} \sim p(s_{t+1}|s_t, a_t)$. And the agent obtains a reward value at each time step $r_t = \mathcal{R}(s_t, a_t)$. The primary goal is to optimize the policy parameters $\theta$ to maximize the reward.

$$\arg \max_{\theta} \mathbb{E}_{(s_t, a_t) \sim p_{\theta}(s_t, a_t)} \left[ \sum_{t=0}^{T-1} \gamma^t r_t \right] \tag{8}$$

where T denotes the time horizon of MDP.

Our ANN teacher policy training process follows (Cheng et al., 2023) to aim for the policy to not directly learn the skills of traversing difficult terrain, but rather to enable the robot to parkour by learning from rewards and following instructions. Thus, unlike approach (Kumar et al., 2021), our method uses privileged information like scandots of terrain, which can be acquired in real-world scenarios, instead of relying on environmental factors like friction. In this phase, the policy externally receives the scandots, and target yaw direction as privileged observations. We utilize various obstacles including gaps, steps, hurdles, and parkour terrain to train the policy.

### 2.4.2 Distilling to SNN

In the initial phase of our training, we employ ANN to create a model capable of generating action and directional commands for executing parkour tasks with quadruped robots. This foundational step establishes the groundwork for our subsequent transition to a more energy-efficient model. We then embark on a distillation process, where the goal is to train an SNN to emulate the decision-making behavior of the ANN. This process begins with the ANN, serving as the teacher network, and interacting with the simulation environment. We proceed by training the SNN, referred to as the student network, aiming to minimize the Mean Squared Error (MSE) loss between the outputs of both the student and teacher networks. To enhance the performance, we further engage the student network in interactions with the environment. During this phase, we continue to measure and minimize the loss under identical environmental conditions. The training process is shown in Figure 3, where the distillation loss are defined as:

$$\mathcal{L}_{action} = \frac{1}{n} \frac{1}{m} \sum_{i=1}^{n} \sum_{j=1}^{m} (action_{ij}^{\text{ANN}} - action_{ij}^{\text{SNN}})^2, \tag{9}$$

$$\mathcal{L}_{yaw} = \frac{1}{m} \sum_{j=1}^{m} (yaw_j^{\text{ANN}} - yaw_j^{\text{SNN}})^2, \tag{10}$$

where $n$ represents the total number of joints in the quadruped robot and $m$ represents the number of training robots. This iterative process of fine-tuning and adjustment enables the SNN to closely match the ANN's output patterns across a variety of scenarios. As a result, the original model's performance is preserved, while the computational load and energy consumption are significantly reduced. This efficiency improvement renders the model more suitable for real-time applications on devices with limited power resources, marking the successful completion of our training process.

Due to the wide dynamic range of event cameras, we can distill our SNN model using models trained under normal lighting conditions with depth cameras, achieving the same effect. This means that although our SNN model is trained with external environmental light different from traditional depth cameras, it can still maintain efficient and accurate inference capabilities under extreme lighting conditions (e.g., direct sunlight or low-light environments). Our approach opens up new possibilities for deploying quadruped robots in more challenging environments, enabling them to perform precise perception and rapid response under almost any lighting condition.

## 2.5 Theoretical Energy Consumption Calculation

To calculate the theoretical energy consumption of SNN, we begin by determining the synaptic operations (SOPs). The SOPs for each block in the spiking model can be calculated using the following equation (Zhou et al., 2022):

$$\text{SOPs}(l) = fr \times T \times \text{FLOPs}(l) \tag{11}$$

where $l$ denotes the block number in the spiking model, $fr$ is the firing rate of the input spike train of the block and $T$ is the time step of the spike neuron. FLOPs($l$) refers to floating point operations of $l$ block, which is the number of multiply-and-accumulate (MAC) operations. And SOPs are the number of spike-based accumulate (AC) operations.

To estimate the theoretical energy consumption of our model, we assume that the MAC and AC operations are 32-bit floating-point implementations in $45nm$ hardware (Horowitz, 2014), with energy costs of $E_{MAC} = 4.6pJ$ and $E_{AC} = 0.9pJ$, respectively. According to (Panda et al., 2020; Yao et al., 2023), the calculation for the theoretical energy consumption of ES-Parkour is given by:

$$
\begin{aligned}
E_{\text{ES-Parkour}} = {} & E_{MAC} \times \text{FLOP}^1_{\text{SNN}_{\text{Conv}}} \\
& + E_{AC} \times \left( \sum_{n=2}^{N} \text{SOP}^n_{\text{SNN}_{\text{Conv}}} + \sum_{m=1}^{M} \text{SOP}^m_{\text{SNN}_{\text{FC}}} \right)
\end{aligned}
\tag{12}
$$

where $N$ and $M$ represent the total number of layers of Conv and FC, $E_{MAC}$ and $E_{AC}$ represent the energy cost of MAC and AC operation, $\text{FLOP}_{\text{SNN}_{\text{Conv}}}$ denotes the FLOPs of the first Conv layer, $\text{SOP}^n_{\text{SNN}_{\text{Conv}}}$ and $\text{SOP}^m_{\text{SNN}_{\text{FC}}}$ are the SOPs of $n^{th}$ Conv and $m^{th}$ FC layer, respectively.

# 3 Experiments

## 3.1 Training Setting

During the transition from ANN to SNN in our distillation process, we embark on an extensive training regimen for the student SNN model. This training is conducted within IsaacGym, utilizing a total of 32 parallel robot simulation environments. These environments are specifically chosen to provide a diverse range of challenges and scenarios, thereby ensuring a comprehensive learning experience for the SNN model. To simulate real-world conditions as closely as possible, we sample event images at a frequency of 10Hz, which allows us to capture dynamic changes within the network inferencing effectively. The training process is powered by an NVIDIA 3090 GPU and spans over a duration of 30 hours. In configuring the SNN, we opt for the Integrate-and-Fire (IF) neuron model, renowned for its simplicity and efficiency. The spiking timestep is set as 4, optimizing the balance between responsiveness and computational demand.

During our training process, we adopt a series of meticulously designed parameters to optimize the performance of our SNN model. Firstly, we set the learning rate to 0.001. For our encoder network structure, we choose the spiking ResNet-18 (Fang et al., 2021a) as our vision backbone. We also use the GRU module (Cho et al., 2014) to fuse the latent features encoded from proprioceptive information and event features. Additionally, we incorporate a 3-layer spiking MLP layer, with sizes [512, 256, 128], to serve as the actor network.

## 3.2 Simulation Results

During the training phase of our quadruped robot's Spiking Neural Network, we closely monitor the terrain level curve to assess the robot's ability to adapt to complex terrains. We evaluate our SNN strategy and the results are shown in Figure 5. This method gradually guides the robot to face tasks of increasing difficulty, significantly enhancing its adaptability and performance under various environmental conditions. It is also important to note that our training achievements are made under varying lighting conditions, which means our curriculum learning can robustly handle changes in lighting.

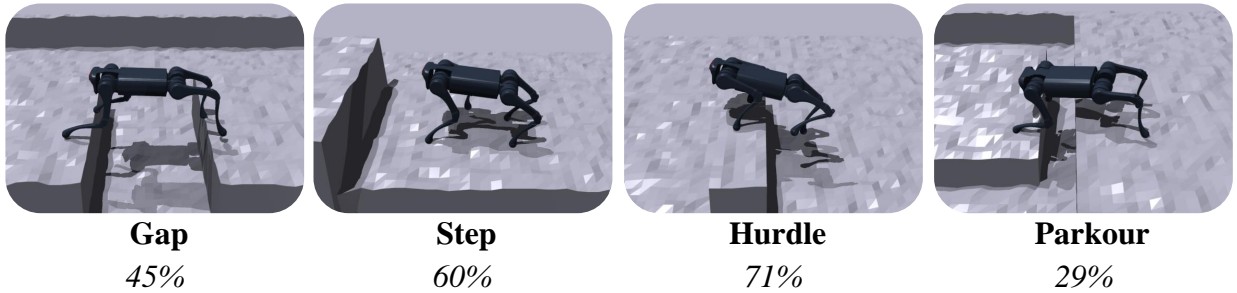

**Gap**
*45%*

**Step**
*60%*

**Hurdle**
*71%*

**Parkour**
*29%*

Figure 5: We evaluate our SNN strategy across four different scenarios. The figure shows the shapes related to each. The top row indicates the type of terrain, while the bottom row displays the success rate for each situation.

|            | Gap    | Step   | Hurdle | Parkour |
|------------|--------|--------|--------|---------|
| ANN        | 808 16 | 876.23 | 853.32 | 1008.6  |
| SNN (ours) | 813.45 | 869.27 | 862.01 | 997.54  |

Table 4: Comparisons of the average robots' joints motor energy (mJ) between ANN and SNN.

| Encoder Type | SNN | | ANN | Efficiency ↓ |
|--------------|-------|------|-------|--------------|
|              | FLOPs | SOPs | FLOPs | OPs(SNN):OPs(ANN) |
| ResNet | $8.00\times e^6$ | $8.76\times e^7$ | $2.04\times e^8$ | **0.46** : 1 |
| MLP    | $7.17\times e^6$ | $2.61\times e^6$ | $3.31\times e^7$ | **0.29** : 1 |

Table 2: Comparisons of the number of operations (FLOPs/SOPs) between the vision encoder of Parkour and ES-Parkour. SNN yields lower operating times than its ANN counterpart.

| Module | Encoder | | Actor |
|--------|-----------------|-------------|--------------|
|        | ResNet (11.19M) | MLP (8.01M) | MLP (0.26M)  |
| ANN Power (mJ) | 0.94 | 0.15 | $1.08e^{-3}$ |
| SNN Power (mJ) | **0.11** | **0.04** | **$3.30e^{-4}$** |
| Energy Saving | 88.29% | 73.33% | 69.44% |

Table 3: Comparisons of Energy Consumption between origin Parkour (ANN model) and ES-Parkour (SNN model). Our ES-Parkour achieves extreme energy saving (up to **88.29%**) in each module.

### 3.3 Analysis of Computing Efficiency

### 3.3.1 Comparisons of the number of operations.

Given that the majority of computational demands in neural networks stem from matrix operations, this section explores the analysis and comparison of the operational counts within the visual encoders of both ANN and SNN. This comparison aims to validate the efficiency of our ES-Parkour system. According to Table 2, the SNN consistently demonstrates a lower total operational count (including both FLOPs and SOPs) compared to the ANN, regardless of whether ResNet or MLP serves as the visual backbone. This difference arises because, the non-spiking portion of the feature (i.e., zero value) in SNNs does not consume computational resources during matrix operations. As a result, the overall number of operations for SNNs significantly falls below that of ANNs. We define an operational efficiency metric:

$$\text{Efficiency} = \frac{\text{OPs(SNN)}}{\text{OPs(ANN)}} = \frac{\text{FLOP}_{\text{SNN}} + \text{SOP}_{\text{SNN}}}{\text{FLOP}_{\text{ANN}}}, \tag{13}$$

where this metric measures the relative energy efficiency ratio, a lower efficiency value (i.e., less than 1) indicates a higher energy efficiency of SNNs compared to ANNs. The reduced efficiency values presented in the table underline the computational efficiency of our ES-Parkour system.

### 3.3.2 Evaluation of the energy consumption.

To further emphasize the low-energy nature of our ES-Parkour, we conduct a detailed comparative analysis of the energy consumption between the proposed ES-Parkour and its corresponding ANN model. As shown

| Scenarios | normal-light | overexposed | underexposed | high-speed |
|---|:---:|:---:|:---:|:---:|
| Anymal parkour (Hoeller et al., 2023) | ✓ | ✓ | ✓ | ✗ |
| Extreme parkour (Cheng et al., 2023) | ✓ | ✗ | ✗ | ✗ |
| Robot parkour (Zhuang et al., 2023) | ✓ | ✗ | ✗ | ✗ |
| **ES-Parkour (ours)** | ✓ | ✓ | ✓ | ✓ |

Table 5: Comparison of the abilities of different methods in extreme scenarios.

in Table 3, using the ResNet scenario as an example, our ES-Parkour presents significantly lower energy consumption, amounting to merely 11.7% of that exhibited by the ANN model. This corresponds to an extreme energy-saving (88.3%) by utilizing SNN. Moreover, the actor module of the ES-Parkour further exemplifies energy conservation compared with ANN Parkour, with $3.30\times e^{-4}$ vs. $1.08\times e^{-3}$, demonstrating the superior low-energy benefits of our systems. We can maintain the same power consumption at the joint level as with ANN, but with better environmental adaptability and lower computational burden as shown in Table 4. Our extensive testing has validated the overall feasibility and performance advantages of the system.

## 4 Conclusion

In this paper, by integrating Spiking Neural Networks (SNN) and event cameras, we not only address the challenges of power consumption and computational load inherent in traditional deep learning models for quadruped robot parkour but also forge a new pathway for enhancing robot perception and control. This approach enables more efficient and adaptive responses in complex environments. We compare the work of robot parkour in Table 5, and our work is the only one that can be tested under all environmental conditions.

Due to the difficulty in obtaining SNN chips, our work has not been tested on actual robots. However, as with previous robotic validation efforts, we have extensively tested our system in simulations to ensure its feasibility on actual robots. This phased research ensures the sustainability of our study. In the future, we will continue to refine our system and advance the integration of the SNN chips with actual robots.

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

## A  Appendix

## B  Video presentation of bio-inspired robot in parkour scenes

In this section, to demonstrate the effectiveness of our bio-inspired robot, we have recorded successful navigation videos[1] of the robot across four distinct scenarios. Each scenario is designed to replicate challenges that may be encountered in natural environments, testing the robot's adaptability, agility, and control. Those scenes are illustrated in Figure 6.

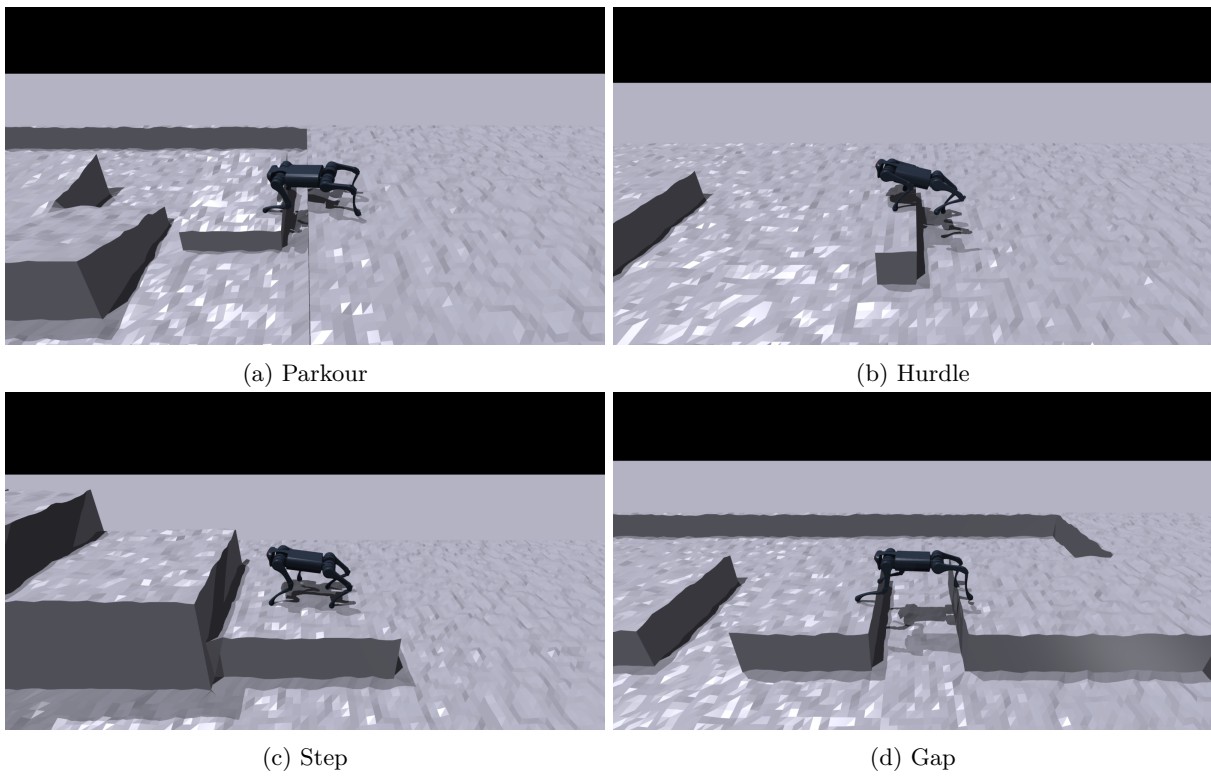

(a) Parkour  (b) Hurdle

(c) Step  (d) Gap

Figure 6: Four challenging parkour scenes.

**Parkour Scenario:** The parkour scenario tests the robot's ability to navigate complex, multi-level environments with precision and speed. It encompasses a series of obstacles that require a combination of jumping, climbing, and balancing maneuvers, showcasing the robot's dynamic movement capabilities and its ability to handle abrupt changes in terrain. (esparkour-parkour.avi)

**Hurdle Scenario:** In the hurdle scenario, the robot is faced with a series of obstacles placed at varying heights and distances. This scenario assesses the robot's jumping accuracy and power, as well as its ability to predict and react to the spacing and height of consecutive hurdles, thereby testing its agility and decision-making in real-time. (esparkour-hurdle.avi)

**Step Scenario:** This scenario involves a series of ascending and descending steps, designed to evaluate the robot's capability in managing changes in elevation. It tests the robot's balance, coordination, and the efficiency of its locomotion algorithms in maintaining stability while navigating steps of varying heights, which mimic the uneven terrains found in natural landscapes. (esparkour-step.avi)

**Gap Scenario:** The gap scenario challenges the robot with discontinuities in the pathway, requiring precise calculation and execution of jumps across varying distances. This tests the robot's ability to gauge the

---

[1]The videos are packaged inside the submitted supplementary materials.

required force and trajectory for successful leaps over gaps, reflecting its spatial awareness and predictive modeling in overcoming obstacles. (esparkour-gap.avi)

Together, these scenarios provide a comprehensive assessment of our bio-inspired robot's performance capabilities in environments that mimic real-world challenges. Through these demonstrations, the robot's advanced design and control strategies are evident, highlighting its potential for diverse applications in complex and dynamic settings.

