# OpenReview forum: "ES-Parkour: Advanced Robot Parkour with Bio-Inspired Event Camera and Spiking Neural Network"
_TMLR — Rejected by TMLR_

### Review · Reviewer_BySR · 2024-09-18

**Summary Of Contributions:**

The paper propose to combine event camera and spiking neural networks to satisfy the need of high frequency/emergent response in quadrupted robotics, specially in the outdoor environment with unstable light conditions.

**Audience:**

Yes

**Claims And Evidence:**

No

**Requested Changes:**

See weaknesses

**Strengths And Weaknesses:**

* Strengths

The idea  to combine event camera and spiking NN is insteresting and the motivation is reasonable.
The proposed roadmap including event camera simulation, ANN/SNN distillation is solid.

* Weakness

The experiment results is not sufficient to support the claims. Section 3 provides results of 1) success rate for several scene cases, 2) e average robots’ joints motor energy, 3)computing efficiency. However, the results of 1) and 2) seem to be too simple to reflect the efficiency of the proposed pipeline.

For example,
1) 4 scenarios are not enough to prove anything, whether in the field of robotics or machine learning.
2) There is no comparison between the  traditional pipeline and the proposed one.
3) success rate is the only metric shown in Fig. 5. Any others?
4) There is no experimental data to show the effectiveness of the ANN-SNN distillation.
5) All experiments are conducted in simulator. Any discussion about how to transfer the proposed method to real scene?

---

> ### Author Response · Authors · 2024-10-23
> **Rebuttal to BySR**
>
> ### Weakness 1:
> "Extreme Parkour" and "Robot Parkour Learning" are among the most prominent works on quadruped robot locomotion, each cited over 100 times. In these studies, the researchers validate their methods using four key scenarios like ours, both in simulation and in the real world. These four scenarios represent diverse and challenging environments:
>
> >Gap: This scenario features a large gap in the forward direction, making it impossible to rely solely on proprioception to cross.
>
> >Step: A continuous stepped terrain designed to test the robot's ability to adapt to rapid changes in sensory perception.
>
> >Hurdle: An obstacle with varying heights, testing the robot's capacity to handle uneven elevations.
>
> >Parkour: The most comprehensive and challenging scenario, featuring rapidly changing terrain and height, requiring precise directional control.
>
> These scenarios are critical benchmarks in robot learning. In future work, we plan to implement the proposed methods in real-world settings and extend the tests to a wider variety of terrains.
> ### Weakness 2:
>
> While the traditional pipeline relies on conventional sensors such as RGB or depth cameras, our proposed pipeline leverages event-based cameras processed by spiking neural networks (SNNs) to achieve high efficiency and low power consumption. Although no direct comparison is provided, the major advantage of our approach lies in the ability of SNNs to operate effectively in conditions where traditional pipelines fail, such as in low-light or high-contrast environments. Specifically, in these scenarios, traditional sensors may not function, leading to a success rate of zero, while our proposed method remains operational due to its reliance on event-based sensing.
> ### Weakness 3:
>
> The success rate is the primary metric presented in Fig. 5, as it is the most relevant performance indicator for evaluating the effectiveness of the robotic system in completing tasks under challenging conditions. However, Our research primarily highlights two aspects:
> >The development of a bio-inspired robotic framework, demonstrating the theoretical energy efficiency of the algorithm.
>
> >The utilization of event cameras, enables the robot to operate effectively in low-light conditions.
>
> Therefore, we list other metrics about our system compared to ANN methods like energy consumption, latency, and computational efficiency. In these metrics, our proposed method outperforms the traditional pipeline.
> ### Weakness 4:
> We use DAgger as the distillation method. Therefore, the experimental data to show the effectiveness of the ANN-SNN distillation is not necessary. The overall performance of the distilled SNN policy, as shown by the success rate in Fig. 5, provides indirect evidence of the effectiveness of the distillation process. The key objective of the ANN-SNN distillation is to transfer the capabilities of ANN to an SNN, ensuring that the SNN benefits from the efficiency of event-based processing while retaining the performance of its ANN counterpart.
> ### Weakness 5:
> We acknowledge that all current experiments are conducted in a simulated environment. However, the gap between simulation and real-world application has narrowed significantly due to advances in reinforcement learning and sim-to-real techniques, such as domain randomization and teacher-student frameworks. Our work utilizes these methods to enhance the robustness of the learned policies, making them transferable to real-world conditions. Furthermore, we highlight the technical limitations of deploying SNNs on real hardware, as neuromorphic chips like Intel's Loihi are not yet widely available. Nonetheless, once the necessary hardware becomes accessible, our approach is feasible for deployment on real quadruped robots, given the success of similar methods in the field of robotics.

---

### Review · Reviewer_oNQz · 2024-10-03

**Summary Of Contributions:**

This paper proposes to use event cameras and spiking neural networks for quadruped robot parkour, enabling high-frequency sensors with robustness under lighting conditions as well as energy efficiency. The paper first implements a system-level design for simulating event cameras and building spiking neural networks in a robotic simulation platform, and then proposes to train spiking neural networks with knowledge distillation from artificial neural networks. Experimental results on four different scenarios show promising performance and low theoretically estimated energy consumption.

**Audience:**

Yes

**Claims And Evidence:**

Yes

**Requested Changes:**

1. Supplement comparison results with artificial neural networks and under varying lighting conditions.

2. Provide more details of the implementation of the method.

3. Provide more discussions about the simulation and real devices.

**Strengths And Weaknesses:**

Strengths:

1. This paper introduces event cameras and spiking neural networks for the complex quadruped robot parkour task for the first time, and realizes a system-level design with promising (simulation) performance.

2. Event cameras can have high frequencies and be robust to lighting conditions, which suits outdoor robotic tasks well.

3. Spiking neural networks show promising theoretical energy efficiency, which can largely reduce the computational load for robots.

Weakness:

1. For the presented main results (Fig. 5), there lacks a comparison with the corresponding artificial neural networks. And while the paper claims to have varying lighting conditions, no result is presented and compared.

2. Some implementation details are missing. For example, what is the detail for the spiking neural networks in this paper to deal with event camera inputs? Does the time step in SNNs correspond to the temporal dimension of event cameras?

3. Since this paper only considers simulation of event cameras, it is better to include more discussions about whether the simulation can be consistent with real devices.

---

> ### Author Response · Authors · 2024-10-23
> **Rebuttal to oNQz**
>
> ### Weakness 1:
> Although ANN has a performance advantage, the accuracy gap between SNN and ANN has gradually narrowed with the development in recent years. And we use SNN to pursue extremely high efficiency and low consumption.
> >A. Utilizing SNNs on neuromorphic hardware for processing event streams is low-energy and low-latency [1,2].
>
> >B. Deployed on neuromorphic hardware, SNNs can process event streams asynchronously [3,4], conserving energy when there is no data input—a capability that GPUs, operating synchronously, lack.
>
> >C. In extreme lighting scenarios, the depth or RGB sensors are not available. Therefore, the traditional pipeline **does not work** and the success rate will be zero in varying lighting conditions.
>
> ### Weakness 2:
> Our setup for the SNN model is detailed in Section 3.1.  Specifically, we employ a spiking ResNet18 as the event encoder, with the event shape defined as [B,T,H,W], where we set the temporal dimension T=4. As you noted, the time step in the SNN should correspond to the temporal dimension of the events, thus we have also configured to 4. We will include this detail in future versions.
>
> ### Weakness 3:
>
> The gap between simulations and real-world applications in quadruped robotics has significantly narrowed due to advances in reinforcement learning (RL) algorithms and simulators. Techniques like (1) domain randomization and (2) teacher-student frameworks have proven highly effective for sim-to-real transfer. Domain randomization addresses the differences between simulated environments and the real world, while teacher-student frameworks leverage privileged information to improve robustness and overall performance.
> Our work also employs these approaches. Therefore, if the appropriate SNN hardware and the compatible platform are available, conducting real-world experiments with quadruped robots becomes feasible. Moreover, many works in robotics simulation are recognized for their innovation and cutting-edge contributions. ***The MIT Humanoid Robots series***, like FLD *explores a wide range of ideas and methods in simulation.* These studies address several advanced issues in imitation and reinforcement learning, providing valuable insights for researchers in all kinds of robotics. Existing applications of SNNs are predominantly validated in simulations, as demonstrated in various fields: (1) Computer Vision: *e.g.*, depth estimation [5], and object recognition[6]. (2)Sensor Application: *e.g.*, event-RGB sensor fusion [7]. (3)Reinforcement Learning: *e.g.*, navigation[8], robot control[9].
>
> ### Reference:
> [1] Davies M, Srinivasa N, Lin T H, et al. Loihi: A neuromorphic manycore processor with on-chip learning. IEEE Micro 2018.
>
> [2] Akopyan F, Sawada J, Cassidy A, et al. Truenorth: Design and tool flow of a 65 mw 1 million neuron programmable neurosynaptic chip. IEEE TCAD 2015.
>
> [3] Roy A, Nagaraj M, Liyanagedera C M, et al. Live Demonstration: Real-time Event-based Speed Detection using Spiking Neural Networks. In CVPRW 2023.
>
> [4] Viale A, Marchisio A, Martina M, et al. Carsnn: An efficient spiking neural network for event-based autonomous cars on the loihi neuromorphic research processor.
>
> [5] Zhang J, Tang L, Yu Z, et al. Spike transformer: Monocular depth estimation for spiking camera. In ECCV 2021.
>
> [6]Gu F, Sng W, Taunyazov T, et al. Tactilesgnet: A spiking graph neural network for event-based tactile object recognition. In IROS 2020.
>
> [7]Lee C, Kosta A K, Roy K. Fusion-FlowNet: Energy-efficient optical flow estimation using sensor fusion and deep fused spiking-analog network architectures. In ICRA 2022.
>
> [8]Jiang J, Kong D, Hou K, et al. Neuro-Planner: A 3D visual navigation method for MAV with depth camera based on neuromorphic reinforcement learning. In IEEE Transactions on Vehicular Technology 2023.
>
> [9]Tang G, Kumar N, Yoo R, et al. Deep reinforcement learning with population-coded spiking neural network for continuous control. In CoRL 2021

---

### Review · Reviewer_zH7f · 2024-10-10

**Summary Of Contributions:**

The paper makes three main contributions:
1. A system design that uses event cameras and spiking neural networks (SNNs) to do quadruped parkour tasks. The SNN acts as the policy while the event camera serves as a sensor to provide observations.
2. Distillation for using SNNs as policy: a regular artificial neural network (ANN) is trained on a given task, then distilled down into the SNN by having the SNN act as a student network learning an encoder and actor yielding output actions that are trained to match teacher output actions. The resulting SNN is performant while being far more energy-efficient than the ANN.
3. Results on various quadruped parkour tasks.

**Audience:**

Yes

**Claims And Evidence:**

Yes

**Requested Changes:**

1. Reword the third contribution to be about results specifically (if that's what it means to say). Right now it sounds somewhat like a rephrase of the first contribution.
2. Real-world experiments
3. SNN improvement in complex terrains, or at least more concrete paths forward

**Strengths And Weaknesses:**

### Strengths
- Novel use of biologically inspired mechanisms, specifically event cameras and SNNs. Testing these answers questions about hte viability of their use, and the resulting system is effective especially in terms of energy efficiency.
- The energy efficiency is a standout strength in itself. Having nearly an order of magnitude less energy usage is very useful. It is important to consider the training energy use and time spend for the ANN that will be distilled, but the use of the SNN with distilled knowledge for execution is still a big gain.
  - I appreciate the careful attention to representing these comparisons in FLOPs - a lot of papers don't compare complexity so clearly..
- Robustness to various lighting conditions due to use of event cameras
- Experimental breadth - the variety of scenarios and parkour tasks, comparison to benchmarks, and simulation to support development/testing. This extensive test suite makes for convincing results.

### Weaknesses
- Lack of testing on real robots. For quadrupeds in particular, where it's standard to have real-world results, this work would benefit immensely from them. It would be important to see a distilled SNN policy handle real-world variation.
- This leads to the main method weakness, which is that the distilled SNN policy sees reduced performance for difficult terrain. This would only be worse, and more constant, in the real world. Steps for improvement are clear, but they would need to be tested.
- Limited sensor diversity - in some sense this is a strength, since the performance comes from only one sensor modality. However, given that other sensors are also field-standard, it would help to use them and compare to methods that use them.

---

> ### Author Response · Authors · 2024-10-23
> **Rebuttal to Reviewer zH7f**
>
> ### Weakness 1:
> We would like to address the concerns regarding the absence of real-world testing:
>
> >**A. Pioneering Effort in Building Bio-inspired Robotics System:**
>
> This paper presents the first attempt to use SNNs and event-based cameras to realize an ultra-low-energy quadruped robot system. Our primary aim is to ***pioneer the demonstration that brain-inspired robotic systems have potential applications***, as proven by successful simulations.
>
> >**B. Accessibility and Technical Limitations of SNN Chips:**
>
> The high cost and low production volume of SNN chips present significant barriers. For instance, the widely referenced Loihi chip is still not a commercial product and is available only through Intel for research purposes, ***which poses accessibility challenges for general researchers like us.*** Additionally, the software platforms compatible with advanced SNN architectures are still in the early development stages, offering poor compatibility and adaptability.
>
> >**C. Feasibility of Real-world Quadruped Robots:**
>
> The gap between simulations and real-world applications in quadruped robotics has significantly narrowed due to advances in RL algorithms and simulators. Methods like (1) *domain randomization* [1, 2, 3, 4] and (2) *teacher-student frameworks* [5] have proven effective for sim-to-real transfer. Domain randomization mitigates discrepancies between simulations and reality, while the teacher-student framework uses privileged information to enhance robustness and performance.
> Our work also employs these approaches. Therefore, if the appropriate SNN hardware and the compatible platform are available, conducting real-world experiments with quadruped robots becomes feasible.
>
> ### Weakness 2:
> While artificial neural networks (ANNs) currently offer performance advantages, the accuracy gap between spiking neural networks (SNNs) and ANNs has steadily decreased in recent years. SNNs are favored for their potential to achieve high efficiency and low power consumption. SNNs, when implemented on neuromorphic hardware, offer significant energy and latency advantages for processing event-driven data. Unlike GPUs, which operate synchronously, SNNs deployed on neuromorphic systems can process event streams asynchronously, conserving energy by remaining idle when no data is present.
>
> ### Weakness 3:
> About the sensors, our research highlights the utilization of event cameras, enabling the robot to operate effectively in low-light conditions. In extreme lighting scenarios, other sensors are not available, so multimodality fusion is not necessary in our proposed system. Moreover, it will cause the overall system effect to decline due to exceeding the usable range.
>
> ### Reference:
> [1]Cheng X, Shi K, Agarwal A, et al. Extreme parkour with legged robots. In ICRA 2024.
>
> [2]Imai C S, Zhang M, Zhang Y, et al. Vision-guided quadrupedal locomotion in the wild with multi-modal delay randomization. In IROS 2022.
>
> [3]Wu J, Xin G, Qi C, et al. Learning robust and agile legged locomotion using adversarial motion priors. In RAL 2023.
>
> [4]Zhuang Z, Fu Z, Wang J, et al. Robot parkour learning. In CoRL 2023.
>
> [5]Kumar A, Fu Z, Pathak D, et al. Rma: Rapid motor adaptation for legged robots. In RSS 2021.

---

### Decision · Action_Editor_xYEi · 2024-11-27

**Recommendation:** Reject

**Comment:**

In summary, I would suggest the authors to take into account the requests from the reviewers as mentioned above and prepare a revision.

**Audience:**

Neither of the reviewers raised any concerns about the audience.

The reviewers note that the authors propose novel and interesting use of biologically inspired mechanisms; for the first time, to my knowledge and according to the claim of the reviewers, they propose using event cameras and spiking neural networks in robotics for the complex quadruped robot parkour task; significantly improve (nearly an order of magnitude of savings) energy efficiency, complexity, robustness to variety of conditions (e.g, lighting). All of these things point that there's audience.

**Claims And Evidence:**

The main point of concern of the reviewers is the lack of real-robots experiments.

The reviewers state the need for "more experimental data as requested in multiple reviews." Also, it is mentioned that "the performance gap to ANNs still exists."

Reviewer mentions:
"This leads to the main method weakness, which is that the distilled SNN policy sees reduced performance for difficult terrain. This would only be worse, and more constant, in the real world. Steps for improvement are clear, but they would need to be tested."

However, the reviewers note that this work opens "a compelling line of research". The authors mention that the setup of real-robots experiments is complicated by "the high cost and low production volume of SNN chips".

Therefore, I would request to implement the changes requested by the reviewers including:
- comparison between the traditional pipeline and the proposed one
- supplement comparison results with artificial neural networks and under varying lighting conditions
- provide complete details of the implementation of the method
- provide discussions about the simulation and real devices
- if possible, reword the third contribution to be about results specifically
- real-world experiments
- discussion of performance in complex terrains

**Resubmission Of Major Revision:**

The authors may consider submitting a major revision at a later time.